# Trends and Challenges in AIoT/IIoT/IoT Implementation

**DOI:** 10.3390/s23115074

**Published:** 2023-05-25

**Authors:** Kun Mean Hou, Xunxing Diao, Hongling Shi, Hao Ding, Haiying Zhou, Christophe de Vaulx

**Affiliations:** 1Université Clermont-Auvergne, CNRS, Mines de Saint-Étienne, Clermont-Auvergne-INP, LIMOS, F-63000 Clermont-Ferrand, France; 2uSuTech Company, 63173 Aubière, France; 3College of Electronics and Information Engineering, South Central Minzu University (SCMZU), Wuhan 430070, China; 4Dong Feng Company, Wuhan 430050, China

**Keywords:** metaverse, digital twin, AIoT platform, TinyML, Zephyr, OpenThread, TensorFlow, Keras, TensorFlow Lite, TensorFlow for microcontroller, neuromorphic computing, IMC, ANN, SNN

## Abstract

For the next coming years, metaverse, digital twin and autonomous vehicle applications are the leading technologies for many complex applications hitherto inaccessible such as health and life sciences, smart home, smart agriculture, smart city, smart car and logistics, Industry 4.0, entertainment (video game) and social media applications, due to recent tremendous developments in process modeling, supercomputing, cloud data analytics (deep learning, etc.), communication network and AIoT/IIoT/IoT technologies. AIoT/IIoT/IoT is a crucial research field because it provides the essential data to fuel metaverse, digital twin, real-time Industry 4.0 and autonomous vehicle applications. However, the science of AIoT is inherently multidisciplinary, and therefore, it is difficult for readers to understand its evolution and impacts. Our main contribution in this article is to analyze and highlight the trends and challenges of the AIoT technology ecosystem including core hardware (MCU, MEMS/NEMS sensors and wireless access medium), core software (operating system and protocol communication stack) and middleware (deep learning on a microcontroller: TinyML). Two low-powered AI technologies emerge: TinyML and neuromorphic computing, but only one AIoT/IIoT/IoT device implementation using TinyML dedicated to strawberry disease detection as a case study. So far, despite the very rapid progress of AIoT/IIoT/IoT technologies, several challenges remain to be overcome such as safety, security, latency, interoperability and reliability of sensor data, which are essential characteristics to meet the requirements of metaverse, digital twin, autonomous vehicle and Industry 4.0. applications.

## 1. Introduction

Metaverse and digital twin are the cutting-edge technologies for the coming year, for many complex applications hitherto inaccessible such as health and life sciences, smart home, smart agriculture, smart city, transport and logistics, Industry 4.0 and social entertainment (video game) applications [1,2,3,4]. Due to the recent tremendous development of process modeling, supercomputing, cloud data analytics (AI, deep learning and neuromorphic computing), communication networks (e.g., 5G/6G, Wi-Fi 6/7 and Bluetooth LE), artificial intelligence of things (AIoT) [5], industrial Internet of Things (IIoT) and the IoT. These disruptive technologies can suppress the barrier between virtual worlds and the real world, and can allow near real-time interaction and decision making. This new paradigm for unification of virtual worlds and real world will drive new applications and practices to optimize (reduce cost) and to improve industry processes and services, i.e., production, maintenance, augmented assistance and tutorial, product lifecycle management and quality.

It is estimated that growth of the IoT (here, the IoT corresponds to the IoE ”Internet of Every Thing”, i.e., smartwatches, smart phones, IIoT/I, etc.) through connected devices across the globe will increase from 7.6 billion to 24.1 billion, with the revenue increasing from USD 465 billion to over USD 1.5 trillion by the year 2030 (according to the IoT total addressable market (TAM) forecast published by Transforma Insights in 19 May 2020) [6].

In general, the IoT platform has three main layers: the IoT cloud server (back-end), edge router and IoT devices (front-end). Due to communication delay between front-end and back-end (upward and downward exchanges), distributed decision making is needed at each layer: IIoT/IoT devices, edge router and the IoT cloud to minimize bandwidth use, and to improve reliability, security and privacy. Moreover, metaverse, digital twin and, in particular, autonomous vehicle applications (dynamic distributed front-end devices) are time sensitive (real-time interaction). Therefore, time-sensitive dynamic IIoT/IoT devices must have the decision-making capability (AIoT device) to interact directly and locally with their environments (e.g., actuators), and to send some defined data to the IoT cloud through an edge router for global decision making.

It is evident that AIoT/IIoT/IoT technology is a crucial research area, and is the cornerstone of metaverse and digital twin as it provides the essential data to power its applications. This work covers the ecosystem of core technologies to implement low-power and low-cost AIoT/IIoT/IoT devices. Here, we address the state-of-the-art hardware (MCU, sensors and wireless carrier), firmware (operating system and communication protocol stack) and embedded deep learning framework (TinyML) by highlighting their trends, challenges and opportunities.

In addition to this introduction, this work is structured as follows: Section 2 describes the current hierarchical AIoT/IIoT/IoT platform and shows its drawbacks for handling time-sensitive applications. Next, in Section 3, we introduce the core technology ecosystem of AIoT/IIoT/IoT devices, and also discuss recent advancements in ANN and SNN that can be used to implement an AIoT device. Section 4 presents the TinyML framework used to implement an AIoT device dedicated to strawberry disease detection. The feasibility of such an AIoT device is validated and evaluated using ST’s B-L4S5I-IOT01A board. The lifetime of the AIoT device implemented using conventional and sub-threshold CMOS technologies is quantified and an evaluation of the TinyML framework is summarized. Finally, we conclude this work and highlight trends and challenges in AIoT/IIoT/IoT technology and the evolution of the embedded deep learning.

## 2. AIoT/IIoT/IoT Platform

An AIoT device can directly interact with other AIoT/IIoT/IoT devices in two ways, i.e., machine-to-machine (M2M) communication, either direct or indirect. Direct M2M communication is performed when the AIoT/IIoT/IoT devices are located in the same wireless coverage area (wireless range). Indirect M2M is performed when the destinations of AIoT/IIoT/IoT devices are out of the wireless range of the source AIoT/IIoT/IoT devices. Therefore, at least one router is needed to forward messages to the destinations of AIoT/IIoT/IoT devices. For example, AIoT/IIoT/IoT devices send messages to the IoT cloud server and the IoT cloud server forwards the messages to the destinations of the AIoT/IIoT/IoT devices, connected to the same IoT platform. Thereby, due to resource constraints, AIoT/IIoT/IoT devices interact with other AIoT/IIoT/IoT devices (out of its wireless range) through at least three routers. First, AIoT/IIoT/IoT devices send messages to the edge router in their wireless range and the edge router forwards the messages to the IoT cloud server. The IoT cloud server forwards the messages to the edge router connected to the destinations of the AIoT/IIoT/IoT devices to deliver the message. This architecture enables use of an appropriate wireless access medium at the front-end layer according to the application requirements (e.g., Bluetooth, Wi-Fi, ZigBee or WMBUS) by considering resource constraints such as available energy, wireless bandwidth and computational resources. Typically, a multi-support edge router (e.g., BLE/ZigBee and 4G/5G or Wi-Fi) is used to link AIoT/IIoT/IoT devices to the IoT cloud server. Thus, the IoT platform has three layers of abstraction, as illustrated in Figure 1: AIoT/IIoT/IoT devices (front-end), edge router and cloud server (back-end). AIoT/IIoT/IoT devices collect real-world data from their environment, interact with their environment (local decision-making to control actuators) and communicate with an edge router or the cloud (e.g., sending environment status). Edge routers or gateways manage, collect and save AIoT/IIoT/IoT sensory data (temporary data), perform real-time data analytics, and communicate and interact with AIoT/IIoT/IoT local devices. Edge routers send defined sensory data and environment status to the IoT cloud server. The IoT cloud server (back-end) performs long-term data analytics (deep learning), long-term data storage, decision making, etc. 

The question is whether this IoT platform can cope with the increasing number of AIoT/IIoT/IoT devices and new emerging time-sensitive (real-time critical interaction) applications such as autonomous vehicle, metaverse, digital twin and Industry 4.0 applications.

P. Ferrari et al. focused on evaluating the communication delay due to data transfer from the source, i.e., from the production line to the IBM Bluemix IoT cloud and the way back. The round trip time (RTT) low average time was less than 300 ms with a standard deviation of about 70 ms and the maximum RTT time was less than 1 s [7]. This RTT was too long for an autonomous vehicle to interact with its environment. It is essential to minimize the RTT; therefore, a smart edge router is developed to offload the cloud server to face the increasing number of AIoT/IIoT/IoT devices and minimize latency. In fact, a smart edge router can drastically reduce the decision-making latency time for a specific application, avoids wasting bandwidth resources (preserves network bandwidth by processing selected data locally) and, to some extent, improves security and protects privacy [8]. However, to improve the IoT platform to meet the requirements of emerging time-sensitive (real-time interaction) security/safety applications such as autonomous vehicles, digital twins and metaverse (AR/VR), it is important to provide IoT devices with local decision-making capability.

## 3. AIoT/IIoT/IoT Device: Hardware and Firmware

### 3.1. State-of-the-Art of AIoT

In fact, nowadays, not all IIoT/IoT devices have the capability to make decisions, low-end IIoT/IoT devices (due to resource constraints) only sample environmental data, send sensory data to the IoT cloud through the edge router and wait for the IoT cloud to decide to perform an action. A High-end IIoT/IoT devices can be integrated with a relatively complicated deep learning algorithm able to perform (intelligent) decision making by interacting directly and locally with defined actuators or neighboring devices. This capability is essential to enable large-scale deployment of IIoT/IoT devices (millions) by avoiding internet network congestion and improving security, safety and privacy. In general, the terms IIoT/IoT do not indicate whether or not the IIoT/IoT devices have cognitive capability.

Hou et al. discussed and highlighted the concept and the difference between a simple/low-end IoT device and a smart object (high-end IoT or AIoT device) [9]. Figure 2 presents a deployment of a basic AIoT platform for a smart irrigation application including local or/and remote servers.

In fact, due to resource constraints, how smart can a smart object be?

The use of artificial intelligence (AI) helps to understand, learn, reason, and interact, and thus, it increases efficiency. AI technology such as machine learning allows many types of correlations of large amounts of structured and unstructured data from multiple sources to extract knowledge and to take action. AI was born officially in 1956, but the first neuron model was defined in 1943 by McCulloch and Pitts [10]. Initially, AI investigated propositional logic and the representation of knowledge, i.e., an expert system [11]. The developed methods and tools were related to knowledge-based reasoning, represented by facts and rules. Over the decades, the field of AI was extended to become a multidisciplinary science such as machine learning, the adaptive control theory, the information theory, and the theory of computation and game theory [12,13]. Deep learning is a subset of machine learning and is a multilayer neural network. Today, deep learning outperforms other algorithms for gesture analysis, video image processing (object detection and recognition) and speech processing (speech recognition and translation). It should be noted that AlphaGo DeepMind of Google is running on a supercomputer built around 1202 CPUs and 176 GPUs [14]. Admittedly, the deep learning algorithms based on von Neumann’s CMOS architecture are efficient, but they consume a lot of energy. For example, the next-generation computer technology is expected to solve problems at the exascale with 10^18^ calculations each second and it will consume between 20 and 30 megawatts of power [15]. Thereby, about 5–15% of the world’s energy is spent in some form of data manipulation, transmission or processing [16]. It should be noted that the human brain only consumes about 20 W. The brain is based on large collections of neurons (approximately 86 billion neurons), each of which has a cell body (soma), an axon and dendrites. The information or action potential is transmitted from one neuron to another neuron through synapses, as illustrated in Figure 3. The neuronal signal action potential or spike consists of short electrical pulses with amplitudes of about 100 mV and typically a duration of 1–2 ms and is generated by the soma as soon as its membrane potential reaches a critical value ϑ [17]. In the 1990s, Carver Mead investigated an unconventional computer architecture for mimicking brain function called neuromorphic computing [18]. Due to the inherent asynchrony in memory computing (synapses) and the sparseness of spike trains, neuromorphic computing is energy efficient for performing cognitive tasks (deep learning algorithms). Note that deep learning is based on multi-layered neural networks and classified into two types: artificial neural networks (ANNs) and spiking neural networks (SNNs). SNNs are considered to be the third generation of neural networks. 

The neuron model of an ANN is based on McCulloch and Pitts model. An artificial neuron takes in some number of inputs (x_1_, x_2_, …, x_n_), each of which is multiplied by a specific weight (w_1_, w_2_, …, w_n)_. The logit of the neuron is $z=(\sum_{i=1}^{n}X_{i}W_{i}+\;b),$ where b is the bias of the neuron. The output of the neuron is expressed as:$$y=f\left(z\right)=f\left(\overset{n}{\underset{i=1}{\sum }}X_{i}W_{i}+\;b\right)\;$$
where *f* is the activation function.

There are four major activation functions: sigmoid, tanh, softmax and restricted linear unit (ReLU). The ReLU activation function is defined as $f\left(z\right)=\max\left(0,z\right)$. In general, to train an ANN, the ReLU activation function, the back propagation algorithm and the stochastic gradient descent (SGD) are used to minimize the ANN output error. The effectiveness of training ANNs with large databases in different applications has been proven; however, this is not the case with SNNs.

Unlike ANNs, it is difficult to train SNNs with direct backpropagation due to the non-differentiable nature of their activation functions. One of the most popular ANN-SNN conversion models used is the linear leaky integrate-and-fire (LIF) model [19]. It is expressed as follows [20]:$$V\left(t\right)=e_{}^{-\frac{t-t_{0}}{\tau_{m}}}\left[\int_{t_{0}}^{t}\frac{I_{inj}\left(t^{\prime }\right)}{C_{m}}e_{}^{\frac{t^{\prime }-t_{0}}{\tau_{m}}}dt^{\prime }+V\left(t_{0}\right)\right]$$
where $\tau_{m}=C_{m}\cdot R_{m}$ is the membrane time constant, I_inj(t)_ is the input current, $C_{m}$ is the membrane capacitor, Rm is the membrane resistor and *V*(*t*_0_) is the initial membrane potential [20].
$$I_{inj}=w_{i}\overset{N}{\underset{j=1}{\sum }}\delta \left(t-j\frac{1}{x_{i}}\right)$$
where $\delta \left(\;\right)$ is the delta function $N=x_{i}\;\mathrm{and}\;T_{w}=\sum I_{inj}$ is the number of spikes in time windows T_w_.
$$\int_{0}^{T_{w}}I_{l}dt=\overset{n^{l}}{\underset{i=1}{\sum }}w_{i}\;.\overset{n}{\underset{j=1}{\sum }}\int_{0}^{T_{w}}\delta (t-j\frac{1}{f_{i}})dt=T.\overset{n^{l}}{\underset{i=1}{\sum }}w_{i}f_{i}$$

In [20], the authors demonstrated the equivalence of a linear LIF/SNN and ReLU-ANN model. This proof allowed ANN-SNN conversion and BP with surrogate gradients and direct supervised learning. Moreover, the leaky integrate-and-fire (LIF) neuron has been implemented with a few arithmetic components, such as an adder and comparator [21]. Kashu Yamazaki et al. presented an overview of different SNN neuron models for different applications [22]. While a spike timing-dependent plasticity (STDP) model, which is biologically plausible allows unsupervised learning, the lack of global information hinders the convergence of large SNNs to be trained with large and complex datasets.

The architecture of neuromorphic computing exploits the property of low-power new nonvolatile memory based on resistive switching materials, such as phase-change memory (PCM), ferroelectric device, valence change memory (VCM), electrochemical metallization cells and spintronics [15], to locally implement the integrate-and-fire function (neuron synapse) in the memory cell, i.e., in-memory computing (IMC). IMC avoids intensive backward and forward data transfers between memory and processing units (CPU, GPU, TPU, etc.) in conventional von Neumann architecture. Consequently, IMC reduces power consumption and throughput latency. The different technologies of new memory materials (memristor) are well described in [15].

**Figure 3 sensors-23-05074-f003:**
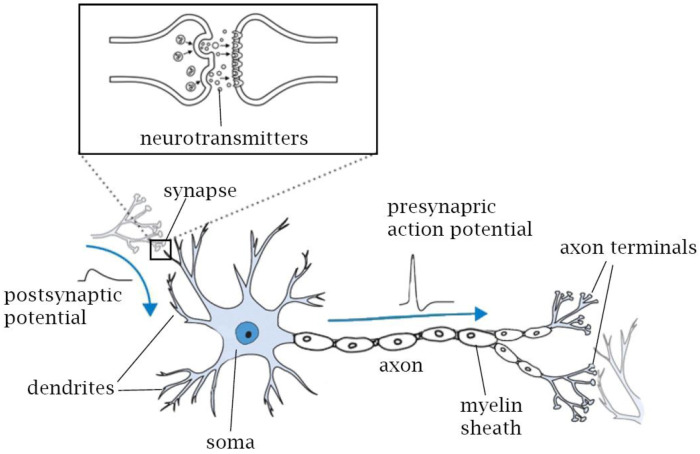
A typical structure of a biological neuron and synapse [22].

Today, there are four categories of neuromorphic computing. The first category (DeepSouth, IBM TrueNorth and Intel Loihi) uses digital CMOS to emulate brain functions. The second category (SpiNNaker, Tianjic, etc.) is based on a software approach (neural network communication protocol and specific hardware architecture) to accelerate ANN and SNN execution. The third category (BrainScaleS, Neurogrid and MNIFAT) uses analogue and mixed-signal CMOS to reproduce a real neuron model [23]. The fourth category uses FPGA-based neuromorphic platforms which outperform previous platforms in terms of power consumption, response time and number of neurons implemented. Yang et al. proposed a CerebelluMorphic system using six Intel Stratix III EP3SL340 FPGAs to realize a large-scale neuromorphic cerebellar network with approximately 3.5 million neurons and 218.3 million synapses [24]. While Wang et al. presented a new abstraction of a neuromorphic architecture into clusters represented by minicolumns and hypercolumns, analogous to the fundamental structural units observed in neurobiology. As a result, implementation on one Altera Stratix V FPGA was able to simulate from 20 million to 2.6 billion leaky-integrate-and-fire (LIF) neurons in real time [25]. The Intel Programmable Solutions Group (neuromorphic computing) and the International Center for Neuromorphic Systems at Western Sidney University (WSU) are building a neuromorphic platform using 168 Intel Stratix 10 PGAs with high-bandwidth memory (HBM) and an accelerator configurable network protocol (COPA) to simulate the human cortex (LIF model). It is estimated that the human cerebral cortex has from 10 to 20 billion neurons and from 60 to 240 trillion synapses [26]. Proof-of-concepts (PoCs) developed on FPGA-based neuromorphic and memristor IMC platforms will lead to the next significant advances in SoC design of low-cost, low-power AIoT devices.

So far, although there has been significant progress in neuromorphic computing over the last decade, its market is insignificant (USD 200 million in 2025) compared to that of conventional computing [15].

Nowadays, there are two emerging trends in AI systems that aim at implementing low-power embedded AI devices: neuromorphic computing and TinyML. In this article, we only focus on the TinyML technology ecosystem for implementing low-cost and low-power AIoT/IIoT/IoT devices and establish its assessment.

Henceforth, it is important to investigate the cognitive techniques that can be embedded to build an AIoT device by considering its resource constraints. The computing power required for an AIoT device is related to the size of the input sensory data, the sampling frequency, i.e., the duty cycle, and the application algorithm. In fact, we can define two categories of AIoT devices: scalar and multimedia. In general, a scalar AIoT device requires less computing power than a multimedia AIoT device and requires less than 0.5 TMAC to run a simple embedded deep learning algorithm (e.g., linear regression). In addition, a multimedia AIoT device requires more computation power to embed deep learning inference according to the applications such as face recognition (0.5–2 TMACs), AR/VR (1–4 TMACs), smart surveillance (2–10 TMACs) and autonomous vehicles (10S–100S TMACs) [27]. Deep learning has the ability to learn without being explicitly programmed, and it outperforms other algorithms for gesture analysis, video image processing (e.g., object detection and recognition) and speech processing (e.g., speech recognition and translation). These functions play key roles in time-sensitive human-machine interface (HMI) requirements for metaverse, digital twin, autonomous vehicle and Industry 4.0 applications. Here, we introduce the available frameworks for developing deep learning applications to be embedded on AIoT/IIoT/IoT devices in Section 4.

### 3.2. AIoT/IIoT/IoT Device Hardware

In general, AIoT/IIoT/IoT devices have four main low-power components:-A processing unit and sensor interfaces based on a single-core or multicore MCU to implement AIoT/IIoT/IoT device safety and security;-Low-cost and low-power consumption sensors usually based on MEMS/NEMS sensors;-A power management unit which is essential to increase the lifetime of the AIoT device by minimizing its power consumption;-Wireless access medium based on single wireless access medium or multiple wireless access medium.

Figure 4 illustrates the basic hardware architecture of AIoT/IIoT/IoT devices.

Due to resource and cost constraints, the design of AIoT/IIoT/IoT devices is usually specific to meet application requirements such as safety, form factor, time sensitive (real-time constraint) and power consumption (lifetime). In fact, resource constraints condition the implementation decision to adopt a specific hardware (MCU and wireless access medium) and firmware architecture (e.g., smartwatch).

Today, for the conventional von Neumann hardware implementation of AIoT/IIoT/IoT devices, two technologies are available: system-on-chip (SoC) and commercial-off-the-shelf (COTS). SoC is used to implement small form factor (e.g., smartwatch), low-power consumption and low-cost (mass production) AIoT/IIoT/IoT devices such as FitBit sense, Apple watch, Huawei watch and medical patches (e.g., Medtronic’s SEEQ cardiac monitoring system), whereas, COTS is applied for small series production, i.e., for testing and validation or more complex AIoT/IIoT/IoT devices. 

Note that the difference between the two approaches is mainly in the use of manufacturing technology because in terms of hardware and firmware architectures, the concepts applied are similar.

#### 3.2.1. Processing Unit of an AIoT/IIoT/IoT Device

Over the past 5 years, IoT and AI technologies have made tremendous progress in the implementation of IoT devices driven by the requirements of the development of smartphones, autonomous vehicles (real-time object detection and recognition), smartwatches and metaverse (VR/AR). The current market trend focuses on implementing high-performance AIoT devices, which require symmetric/asymmetric multicore architecture for fault tolerance and computing power (deep learning). MCU manufactures such as ST and NXP provide MCU target portfolios based on ARM IP (e.g., ARM-M and ARM-A) for each application domain: smartwatch, smart care, smart home, smart car and factory automation. Note that, nowadays, ARM MCU IP dominates the embedded market, but this dominance will be reduced in the coming year due to the open-source RISC-V IP. By 2025, 40% of application-specific integrated circuits (ASICs) will be designed by OEMs, up from around 30% today [28].

New emerging microcontroller manufacturers such as Espressif [29], STARFive Technology [30] and ANDES Technology [31] are offering new microcontroller portfolios dedicated to implementing low-cost AIoT devices based on open RISC-V (open source IP). Moreover, new MCUs based on subthreshold technology (e.g., Ambiq subthreshold power optimized technology (SPOT)) and MRAM significantly increase AIoT device resources (memory and computing power) while minimizing power consumption (e.g., Apollo4 from Ambiq) [32]. In addition, today, co-design SoC CAD tools (Figure 5) such as Cadence provide a wide choice of processing units from single core to symmetric, specific symmetric (DSP) and asymmetric multicore to ease the SoC implementation of AIoT/IIoT/IoT devices [33].

#### 3.2.2. AIoT/IIoT/IoT Wireless Access Medium

A wireless access medium is the critical component of AIoT/IIoT/IoT devices in terms of power consumption, data reliability and latency. Many advances in the field have been achieved but it is still not sufficient to meet their widespread use in terms of safety (reliability) and security, communication range and bandwidth, power consumption and interoperability, despite the many IEEE standards available. Therefore, new low-energy and high-bandwidth wireless access media, i.e., IEEE 802.11ah (Wi-Fi 6 and Wi-Fi HaLow) and BLE 5.x. are being developed to meet different domain applications such as VR and AR. Wi-Fi and Bluetooth (classic and low energy) are set to dominate throughout the forecast period with almost three-quarters of connected devices using these technologies due to smart phones and smartwatches, and, as a result, local wireless communication represents 53% of the wireless access medium, according to the review in [34]. 

In Table 1, the Tx max current is indicated, knowing that wireless power consumption depends on the duty cycle and message size of the AIoT/IIoT/IoT application.

Note that Bluetooth and Wi-Fi are the most used wireless access media to implement smart object devices driven by smartphones and smartwatches. Hence, the IoT ecosystem, in particular, the wireless access medium continues to evolve quickly and is steadily driven by 5G/6G and Wi-Fi 6/7.

#### 3.2.3. AIoT/IIoT/IoT Sensor

As wireless access media, embedded sensors constitute the key components of AIoT/IIoT/IoT devices and significantly impact their form factor and power consumption (lifetime). According to a recent Allied Market Research report, the global smart sensor market will grow at a compound annual growth rate (CAGR) of 18.6 percent or USD 143.65 billion by 2027 (from 2020). A sensor quantifies the measurand of the surrounding environment and provides an electrical output signal corresponding to the amount of measurands present. There are different classifications of sensor devices [45]. Table 2 presents a sensor classification by considering their signal stimulus type and the measurand (attribute).

Nowadays, different technologies are used to implement different types of sensors, but from our point of view, MEMS/NEMS technologies are the most promising for implementing many types of sensors with low-cost, low-power consumption and small form factor for various applications from smart wearable sensors to Industry 4.0. In fact, MEMS/NEMS technologies can be used to implement low-cost sensors to measure stimulation signals, as shown in Table 2. These sensors can detect a wide spectrum of signals from the physical world. Due to economic and technological challenges, MEMS/NEMS technologies continue to make very rapid and significant progress [46]. CMOS and MEMS/NEMS technologies will be more and more tightly coupled, which will enable implemention of SoC smart, low-cost and low-power consumption smart object devices [47]. In the near future, CMOS VLSI circuit, MEMS/NEMS sensors (Figure 6a) [47] and CMOS-compatible MEMS/NEMS switch (Figure 6b) [48] may be implemented on the same substrate. These significant achievements will reduce the power consumption, cost and form factor of AIoT/IIoT/IoT devices. In fact, on the one hand, it will allow removal of the interface between the sensors and the MCU, and on the other hand, the MCU will only be activated (powered on or woken up) via a CMOS compatible MEMS/NEMS switch when a relevant sensory input has occurred.

### 3.3. AIoT/IIoT/IoT Firmware

#### 3.3.1. Operating System: Requirements, Trends and Challenges

There are three classes of IIoT/IoT devices: high-end, middle-end and low-end. The high-end AIoT/IIoT/IoT devices such as smartphones have the necessary resources to run ANDROID, Linux or iOS (>32 GB of flash memory and >2 GB of DRAM). The middle-end AIoT/IIoT/IoT devices are based on multicore MCU (at least dual core) and have more than 1 MB of flash and RAM memory (e.g., ESP-EYE), while, in general, the low-end AIoT/IIoT/IoT devices have less than 1 MB of flash memory and 48 KB of RAM (e.g., smart lamp).

Thereby, since the conception of the IoT concept in 1999 [49], many works have been conducted world-wide to develop new operating systems dedicated to resource constraint AIoT/IIoT/IoT devices: Contiki, TinyOS, LiveOS, Mantis, FreeRTOS [50,51] and Zephyr [52]. Native Contiki and TinyOS adopt the event-driven concept. An event is interruptible but not pre-emptible. As a result, a running event is continued to completion. Therefore, event-driven OS is not appropriate for real-time applications. Two concepts may be used to implement a real-time multitask operating system: monolithic and microkernel concepts. Mantis is a multithreaded and monolithic operating system, where FreeRTOS and Zephyr are based on the microkernel concept. In [53], the advantages and drawbacks of these two concepts are presented. Each task or thread has an assigned priority and, according to its priority level, an executing task may be pre-empted when a higher priority task is ready. Thus, a multitask operating system is appropriate for real-time applications to meet time-sensitive application requirements. TOSThread (modified versions of TinyOS) and Contiki multithread support the multitask operating system concept, developed complementarily on the basis of the event-driven kernels [50]. LiveOS is a native hybrid event-driven and multitask operating system. LiveOS has two schedulers: a multithreaded scheduler and an event-driven scheduler. Tasks can be classified into two types: RT tasks and non-RT tasks. RT tasks are pre-empted and are scheduled by the multithreaded scheduler. Non-RT tasks have less time constraints, and they can be scheduled by the event-driven scheduler. With the hybrid scheduling mechanism, preemption will no longer be performed among the non-RT tasks. Consequently, the thread switch frequency is decreased, and then the stack-shifting frequency can be reduced. By doing this, the LiveOS dynamic-stack scheduling performance can be improved [50]. FreeRTOS is based on a microkernel architecture and adopts a multithreading approach where each thread or process can be interrupted and be pre-empted to execute higher priority process.

Zephyr is a multithreading operating system based on a microkernel and nanokernel architecture. Therefore, Zephyr is more flexible than FreeRTOS because it may be configured to run nanokernel to meet low-end IIoT/IoT devices (Figure 7).

Today, FreeRTOS is the most used operating system for AIoT/IIoT/IoT devices and it is integrated into the middleware software package by most MCU manufacturers: ST, NXP, Silicon Labs, Renesas, Espressif, etc. The minimum memory resource requirements are 4 KB of RAM and 9 KB of ROM. More details of resource requirement are provided in [54]. Additionally, AWS facilitates IoT cloud integration for smart objects running FreeRTOS [55,56].

FreeRTOS and Zephyr operating systems support security and safety features. Furthermore, a microkernel architecture is more appropriate to manage processing units based on a multicore architecture, and it can be configured to adapt to the safety application requirements (fault tolerance).

#### 3.3.2. Communication Protocol Stack

Nowadays, the interoperability of AIoT/IIoT/IoT devices is always a challenge. Interoperability of AIoT/IIoT/IoT devices is essential for their development and widespread worldwide deployment. To make interoperability effective, three main components must be specified, standardized and adopted: wireless physical layer (e.g., BLE 5.x and IEEE802.15.4), communication protocol stack (e.g., OpenThread [57,58]) and application layer protocol (e.g., Matter protocol [59]). It is a beginning, big IT players such as Google, Apple, Amazon and Huawei have adopted OpenThread and Matter protocol for smart home applications.

OpenThread is an open-source, official/complete implementation of Thread. Kim et al. provided a good comparison between the IPv6 routing protocol for low-power and lossy networks (RPL) [57] and Thread. It is a complete network implementation including all network layers, and it is developed to meet the following requirements: security, scalability, resilience, energy efficiency and IP-based network [58]. OpenThread adopts two-tiered mesh topology to comply with the no single point of failure principle (Figure 8a). A Thread network is composed of border routers (at least two) and Thread devices (up to 511). The relationship between Thread devices is based on the parent–child relationship. 

There are two types of Thread devices (Figure 8b): full Thread devices (FTDs) and minimal Thread devices (MTDs). The radio of an FTD is always on (ready to act to minimize latency time). FTDs are classified into three categories: full end devices (FEDs), routers (up to 32) and router eligible end devices (REEDs). An MTD sends all the messages to the parent device and does not have the ability to multicast router address. MTDs are classified into two categories: minimal end devices (MEDs) and sleepy end devices (SEDs).

A border router is the bridge between a Thread network and a non-Thread network and it configures external connectivity. A Thread leader is a dynamically self-elected router that manages routers, aggregates and distributes network configuration. It should be noted that the Thread router is elected as the leader without requiring user intervention.

Matter is the successor of the connected home over IP (CHIP) project [59], and is an application layer that can unify devices that are operating under various IP protocols, allowing them to communicate across platforms (Amazon’s Alexa, Apple’s HomeKit, Google’s smart home ecosystems, etc.). Matter has been developed with the following goals and principles in mind [59]:Unifying: Matter will produce a new specification, built on existing market-tested technologies.Interoperable: The specification permits communication between any Matter-certified device, subject to user authorization.Secure: The specification takes advantage of modern security practices and protocols.User control: The end user controls permission to interact with devices.Federated: No single entity serves as a regulator or single point of failure for the root of trust.Robust: The set of protocols specify a complete lifecycle of a device—starting from the seamless out-of-box experience, through operational protocols, to device and system management specifications required for proper function in the presence of change.Low overhead: The protocols are practically implementable on low computing resource devices, such as MCUs.Low memory footprint IP protocol: The protocols are widely usable and deployable.Ecosystem-Flexible: The protocol must be flexible enough to accommodate deployment in ecosystems with different policies.Easy to use: The protocol should aim to provide smooth, cohesive, integrated provisioning and out-of-box experiences.Open: The Project’s design and technical processes should be open and transparent to the general public, including to non-members wherever it is possible.

## 4. AIoT/IIoT/IoT Device Use Case

### 4.1. TinyML Framework

Today, many deep learning open source frameworks are available to ease the development of AI applications for server or cloud servers running on supercomputers (unlimited computational resources and memory) based on general purpose CPUs, tensor processing units (TPUs) and graphic processing units (GPUs): PyTorch (Meta), TensorFlow (Google), SageMaker (Amazon), Azure Machine Learning (Microsoft), Core ML (Apple), Caffe (BAIR UC Berkeley), etc. These frameworks are a valuable source of workloads for hardware researchers to explore hardware-software trade-offs.

Since 2019, an emerging technology, named tiny machine learning (TinyML), has been addressing the challenges in designing power-efficient (mW range and below), deep learning models, to be integrated into embedded systems such as AIoT/IIoT/IoT devices [60].

Figure 9 shows the deep learning algorithm development workflow: First, the dataset should be built according to the adopted neural network model (CNN, RNN, etc.). In some cases, to optimize the training step and improve performance, dataset preprocessing maybe needed. The training step is performed on a work station or supercomputer server by using TensorFlow or another framework (e.g., PyTorch). In general, the precision of the trained model is float32 and the required memory footprint for its parameters are important (greater than MCU available flash and SRAM). In the TensorFlow framework, TensorFlow Lite provides a converter to optimize the trained model in order to reduce the memory footprint and computation power (int8 instate of float32).

From the obtained training model (the model saved in the xx.h5 file), the TensorFlow Lite framework converter generates three optimized models: int8 quantization, integer dynamic quantization and float16. One of the three files (e.g., int8 quantization tflite file) can be used in another tool such as the X-CUBE-AI [61] extension pack of STM32CubeMx [62] to generate C codes to be implemented on the ST MCU portfolio. Similar to the TensorFlow for the microcontroller framework, the Apache micro tensor virtual machine (µTVM) deep learning framework is based on two main components (Figure 10): µTVM runtime and AutoTVM. AutoTVM extracts operator requirements automatically from a neural network computation graph, and then leverages a machine learning guided optimizer to tune operator implementations for performance on real hardware. The µTVM runtime bloc is mainly a cross compiler to generate binary code to be executed by a target MCU [63]. The authors show that µTVM enables development of ML applications to be embedded on bare-metal devices such as STM Nucleo-F746ZG, STM STM32F746 Discovery and nRF 5340 Development Kit [63,64].

It should be noted that, in general, MCU manufacturers such as ST (X-CUBE-AI), DXP (eIQ ML) and ARM (ARM NN), provide TinyML tools that accept TensorFlow Lite converter file (tflite) to target their MCU portfolios.

Figure 11 presents the workflow using the TensorFlow framework to train and validate a deep learning model and to convert the validated model by using a TensorFlow Lite converter: Integer 8-bit Quantized TFLite Model. The generated tflite model is evaluated by using the X-CUBE-AI expansion pack of STM32CubeMX to target the ST MCU portfolio.

### 4.2. Feasibility of Implementing an AIoT Device for Plant Disease Detection

Many works have been carried out worldwide by using deep learning algorithms in order to improve crop yield and to minimize fresh water use in different agriculture domains: disease detection, plant classification, land cover identification and precision livestock farming [65]. Li et al. reviewed the research progress over the past few years, focusing on deep learning models used to detect plant leaf disease identification [66]. Pandian et al. proposed a novel 14-layer deep convolutional neural network (14-DCNN) dedicated to plant disease detections (aloe vera, banana, apple, etc.). 14-DCNN was developed by using the TensorFlow framework and ran on an HP Z240 for prediction. On the 8850 test images, the proposed DCNN model achieved 99.9655% overall classification accuracy. 14-DCNN required 17,928,571 parameters and 37 MB of memory [67]. The obtained results are promising but the necessary resources are too important to be embedded with the same algorithm on AIoT/IIoT/IoT devices. Therefore, how to implement an efficient DL algorithm to be embedded into an AIoT device for plant disease detection is a challenge.

Nowadays, to the best of our knowledge, AIoT devices dedicated to the detection of plant diseases and pests are rare. In this paper, we show that a CNN deep learning algorithm dedicated to plant disease detection may be embedded into high-end low-cost and low-power IoT devices such as a STL32L4S5i discovery kit (B-L4S5I-IOT01A board) [68] and an ESP-EYE development board [69]. Since AIoT devices have limited computing power and memory resources, our approach is to implement a deep learning algorithm to detect diseases of one species of plant at a time, such as strawberry (healthy or scorch). When the strawberry leaf scorch is detected, its image is sent to the edge router or the IoT cloud server to diagnose the strawberry disease. The development, training and validation process is as follows: Anaconda navigator 2.3.2, Spyder 5.2.2 and Python 3.9.12 are used to run TensorFlow Keras framework version 2.9.1. on a Lenovo ThinkPad X1 laptop. To train the CNN deep learning algorithm dedicated to strawberry disease detection (healthy or scorch), a strawberry Mendeley dataset containing 1000 images of healthy leaves and 1119 images of scorched leaves [70] is used. Figure 12a shows a sample of three healthy strawberry leaves and Figure 12b shows a sample of three scorched strawberry leaves.Hyper parameters such as batch, epoch and number of CNN units and layers are adjusted to obtain the best prediction results (Table 3).A TensorFlow Lite converter is used to convert the float32 model into float16 quantization, 8-bit integer quantization, and integer dynamic quantization. The converted models are validated, evaluated and embedded into the B-L4S5I-IOT01A board.An Int8 quantization tflite file is used to target a MCU. In this paper, we use the X-CUBE-AI expansion pack of STM32CubeMX and STM32CubeIDE [71] to evaluate and validate the trained model. It should be noted that most deep learning tools of MCU manufacturers such as ARM and DXP accept tflite model file to target their MCU portfolios. Moreover, Keras saved model file (xx.h5) may also be used. All the CNN network parameters generated from xx.h5 use float32 precision, i.e., an FPU is needed to speed up real-time inference. Therefore, from the Keras xx.h5 model, the generated codes and data will consume at least four times more memory than the int8 quantization tflite file.It is important to note that our CNN model may be retrained to detect the diseases of other plants such as tomato and potato diseases.

The training parameters are initialized with the following values: seed = 256, epochs = 15 and batch size = 16.

Table 3 shows the accuracy results of the four models: baseline Keras, float16 quantization, int8 quantization and dynamic quantization tflite, according to the shape of the CNN. 512 > 64 > 2 corresponds to the dense three-layer CNN, as illustrated by the following codes:x = Dense(512, activation = ‘relu’)(resnet.output)x = BatchNormalization()(x)x = Dense(64, activation = ‘relu’)(x)x = Dropout(0.2)(x)predictions = Dense(2, activation = ‘softmax’)(x).512 > 64 > 2 (578 neurons) provides the best prediction (99.2%) for float32 precision and the worst prediction for int8 quantization tflite model (85.7%).

However, the dense four-layer CNN (128 > 64 > 32 > 2, 226 neurons) outperforms the dense three-layer CNN (578 neurons) for the int8 quantization tflite model with 96.4% prediction accuracy and for the number of neurons used (the loss of precision is 1.2% for the int8 quantization tflite model).

Figure 13 and Figure 14 show that the memory resource requirements of this model are:Flash memory, 87.32 KB;SRAM, 18.47 KB.

Its complexity is 645,952 multiply accumulation complexity (MACC); therefore, an inference can be done every 8.07 ms (1 MACC per clock cycle, 80 MHz MCU). Therefore, our model can perform real-time strawberry diseases detection with a 50-fps video color camera having 128 × 128 24-bit resolution.

Due to its low resource requirement, our model can also be implemented on ESP-EYE and Sparkfun Edge Apollo3 Blue development boards [69,72].

According to the desktop and target validation results, the accuracy of the model does not change when the model is executed on the STM32B-L4S5i board, the difference between the desktop l2r and the target l2r is near zero (0.00000032). It should be noted that STM32CubeMx generates the model C code to be built and debugged on the STM32CubeIDE, as shown in Figure 15. The input image files of the inference program running on the STM32B-L4S5i board are manually downloaded one by one through the serial port using the Ymodem protocol. Figure 16 shows the results of the inference running on the STL32B-L4S5i discovery kit. The obtained results validate the feasibility of the AIoT device based on the STM32L4 microcontroller portfolio. However, the accuracy of strawberry diseases detection still needs to be validated in cultivated fields where the environment is continuously changing. Online training can continuously improve detection accuracy, but the lack of resources (computing power and memory) is the main obstacle. One of the solutions is that the CNN training model can be updated remotely, taking into account the evolution of the environment. Another solution is to implement an unsupervised learning CNN algorithm on the target MCU.

Our CNN model requires only 18.47 KB of RAM and 87.32 KB of flash memory. Therefore, roughly, a low-cost and low-power AIoT device can be implemented using only four main COTS components, as shown in Figure 17.

The AIoT device dedicated to detect plant disease has:One low-cost and low-power (70 mW) CMOS USB camera (CMT-03MP-GC0308-H455) is used to sample strawberry leaves and send its images to the MCU through USB port [73];One low-power MCU is used to run the deep learning algorithm;One BLE controller, which is connected to the MCU through the SPI interface and is used to send the detection result to the edge router;One TI TPS61030 (boost DC/DC converter), with 95% efficiency and adjustable voltage output (1.8–5.5 V) [74], is used to generate the power supply with two standard AA batteries (1.5 V), each having a capacity of 2500 mAh.

To evaluate the power consumption of the proposed AIoT device, two MCU portfolios, ST and Ambiq, are used. In continuous active mode, the maximum power consumption of each component is shown in Table 4. 

To calculate the power consumption of the AIoT device, Formula (1) is applied.
(1)$$\varphi_{t}=\sum_{i=1}^{N}\varphi_{i}$$
where *N* (*N* = 3) is the total number of components used. The power consumption of each component $\varphi_{i}\;$ is: (2)$$\varphi_{i}=U\times I\;or\;\frac{I}{1\mathrm{MHz}}\times U\times MCU_{Frequency}\;$$
where U is the supply voltage (V), *I* is the current, and *MCU_frequency_* is the operating frequency of the MCU. The power consumption of the AIoT device, based on the ST MCU portfolio $\varphi_{t}$, is equal to (39.6 + 24.6 + 70) = 134.2 mW.

The power consumption of the AIoT device, based on the Ambiq MCU portfolio, $\varphi_{t}$, is equal to (1.32 + 9 + 70) = 80.32 mW.

In continuous active running mode, the lifetime of the AIoT device is calculated by applying Formula (3):(3)$$LifeTime\left(h\right)=\frac{Battery\;Capacity\;\left(\mathrm{mAh}\right)\times N\times Battery\;output\;Voltage\;\left(V\right)\times DCDCconverter\;efficiency}{Power\;consumption\;of\;AIoT\;device\;\left(\mathrm{mW}\right)}\;$$
where *N* is the battery numbers.

The lifetimes of the two AIoT devices are calculated and their power consumption is compared with that of the HPZ240 tower PC (65 W), as illustrated in Table 5 [67].

Note that the lifetime of AIoT device can be improved significantly by adopting sleep-wakeup technique as it consumes less power in sleep mode (Table 6). In fact, the evolution of plant diseases is slow; therefore, to monitor plant diseases, the sleep-wakeup technique can be adopted. In this case study, eight samples in daytime (from 9 h to 17 h, one sample every hour) are appropriate to detect plant disease. 

The CMOS camera consumes 70 mW representing 52% of the total power consumption if the AIoT device is implemented using the ST MCU portfolio. Thus, to overcome this problem, an analog switch (LTC1470/LTC1471 of linear technology is used to switch-off the CMOS camera during the sleep period of the AIoT device (Figure 18). The plant disease detection application workflow during the active period is:Sample an image (128 × 128 24-bit color image), it will take 20 ms (50 fps);Run inference to detect strawberry disease: scorch or healthy (5.38 ms);Send a short message or leaf scorch;Raw image (128 × 128 × 24) to the edge router through the BLE controller (393.21 ms = 128 × 128 × 24/1 MBPS).

The total runtime required for the application is 418.6 ms. Notice that we place AIoT in the worst case by considering that during the active period all the 3 components are active during 1 s and the raw strawberry image is sent to the edge router.
$$\varepsilon_{i}^{active}=\varphi_{i}^{active}\times t_{i}\;where\;\varphi^{active}=\overset{N}{\underset{i=1}{\sum }}\varphi_{i}^{activer}\;and\;t_{i}\;is\;the\;execution\;time\;$$
$$\varphi_{i}^{sleep}=I_{i}^{Q}\times V_{i}\;or\;\varphi_{i}^{sleep}=I_{i}^{sleep}\times V_{i}$$
where
$$\varepsilon_{i}^{sleep}=\sum_{i=1}^{N}\varphi_{i}^{sleep}\times t_{i}\;where\;\varphi_{i}^{sleep}=I_{i}^{Q}\times V_{i}\;or\;\varphi_{i}^{sleep}=I_{i}^{sleep}\times V_{i}\;\mathrm{where}\;\mathrm{and}\;t_{i}{\;\mathrm{is}\;\mathrm{the}\;\mathrm{sleep}\;\mathrm{time}\;}_{}^{}$$

$I_{i}^{Q}$ is the quiescent current (e.g., quiescent current of the boost DC/DC converter) and $I_{i}^{sleep}$ is the sleep current of the component i (e.g., sleep current of the MCU).

To give an idea of the lifetime of the AIoT device, each time it wakes up, the active time of the camera, MCU and BLE controller is 1 s. Therefore, per day, the total active time is 8 s and the rest of the time is sleep time.
(4)$$Lifetime\left(d\right)=\frac{\left(Battery\;capacity\;\left(\mathrm{mAh}\right)\times Battery\;voltage\left(V\right)\times Battery\;number\right)\times DCDC-efficiency}{\left(\varepsilon^{sleep}+\varepsilon^{active}\right)\times 24\;h\;\left(J\right)}$$

By applying Formula (4), for the Ambiq MCU portfolio, the MCU is set to deep sleep 3 and the BLE, the lifetime of the AIoT device is 226 days (0.62 year).

For the ST MCU portfolio, the MCU is set to RTC clocked by LES quartz in low drive mode and the BLE controller is set to deep sleep mode, its lifetime is 303 days (0.83 year). Note that the lifetime of the AIoT device can be further improved by activating each component just for the time needed, for example, the camera is switched on only 21 ms instead of 1 s.

In this case study, we show that the lifetime of the AIoT device using the ST MCU portfolio based on classic CMOS technology, due to its very low deep sleep current, outperforms the Ambiq MCU portfolio based on subthreshold technology. Therefore, according to the AIoT device application running mode and the time constraints, its lifetime can be optimized by choosing appropriate MCU technology.

### 4.3. Assessment of TinyML Framework

TinyML and neuromorphic computing are both emerging technologies that aim at minimizing power consumption and improving neural network algorithms execution efficiency (ANN and SNN). However, their approaches are different: TinyML follows the ANN based on von Neumann architecture, whereas neuromorphic computing imitates the brain functioning by adopting nonconventional von Neumann architecture based on new material such as memristor. We showed in the previous section that the TinyML framework, with an appropriate hardware architecture, can be used to implement a low-cost and low-power AIoT device dedicated to strawberry disease detection with acceptable accuracy, a result that has not bene achieved before. Indeed, some ultra-low-power neuromorphic-embedded devices based on an analog VLSI circuit, such as silicon retina [18] and an implantable cardioverter defibrillator (ICD) system, have been implemented [19]. However, for their large-scale implementation, analog circuits’ variability and precision must be overcome [15]. Moreover, due to the non-differentiable nature of SNN activation functions, it is difficult to train SNNs with direct backpropagation, and BP with surrogate gradient performance is behind conventional ANNs [79]. In addition, nowadays, there are few open SNN frameworks compared to those of the ANN. Furthermore, new skills are required to effectively exploit neuromorphic computing based on non-conventional von Neumann architecture in order to develop and train SNN models. For all these reasons, today, TinyML is more popular and easy to use because its development ecosystem is mature and available (open source). Furthermore, TinyML is the extension of well understood and mastered ANN frameworks. Nevertheless, work in neuromorphic computing will lead to the development of new materials such as low-power non-volatile memory and unsupervised learning algorithms. We believe that the next breakthrough technology, will result from cross-fertilization of ANN and SNN ecosystems. We plan to study a hybrid neural network (HNN) that combines input layers (SNN) based on MEMS/NEMS sensors and a new non-volatile memory to emulate neuronal synapses (IMC), and hidden and output layers based on ANN. Furthermore, advanced unsupervised ANN and SNN algorithms are an important lever for the development of distributed AIoT devices. Finally, due to the inherent multidisciplinarity of this disruptive new technology, teamwork is also a key element to drive the development and successfully implement of the next-generation AIoT devices.

## 5. Conclusions and Perspectives

Smart everything is under way, driven by the development of digital twins, metaverse, Industry 4.0 and autonomous vehicles, and it has enjoyed an incredible synergy of advancements in its core hardware and software technologies such as firmware (Zephyr and OpenThread), deep learning for microcontroller tools (e.g., X-CUBE-AI), VLSI and MCU (ultra-low power subthreshold and new memory technologies and new asymmetric multicore RISC-V architecture), MEMS/NEMS (fusion of MEMS/NEMS and CMOS processes) and wireless technologies (e.g., low-energy high-bandwidth and medium-range Wi-Fi Halow). These advances enable the implementation of low-cost, low-power AIoT devices for Industry 4.0, smart homes, etc.

Moreover, the interoperability of AIoT devices will soon be solved (e.g., Matter for smart home). These news trends will allow development of many applications hitherto inaccessible such as health and life sciences, smart home, smart agriculture, smart city, transport (autonomous vehicle) and logistics, Industry 4.0, entertainment (video game) and social media applications.

Our future work will focus on an evaluation of the accuracy of field-embedded real-time plant disease detection according to the different tflite files (i.e., int8, integer dynamic quantization and float16) and color camera resolutions.

In fact, the current plant disease detection application code is a bare-metal code, where, in general, an application runtime error occurrence is fatal (failure cannot be recovered). It should be noted that, nowadays, a TinyML framework to target specific asymmetric multicore microcontrollers is not yet available (lack of optimized libraries and converters). Consequently, it is not easy for a designer to implement high-performance AIoT devices. Moreover, we will investigate a hybrid neural network (HNN) to implement high-performance, low-power and low-cost AIoT devices.

## Figures and Tables

**Figure 1 sensors-23-05074-f001:**
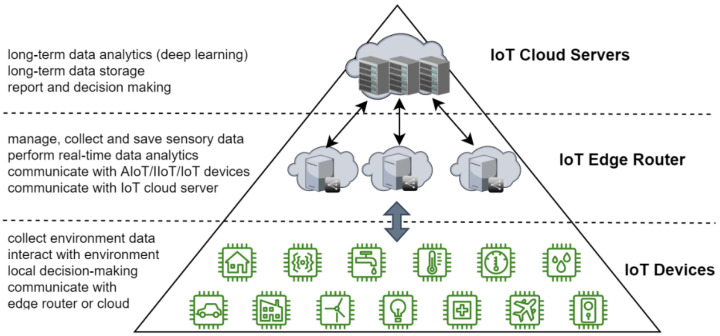
IoT platform.

**Figure 2 sensors-23-05074-f002:**
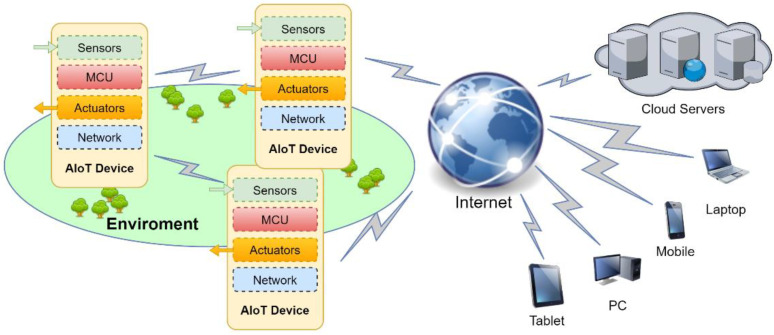
Basic AIoT platform.

**Figure 4 sensors-23-05074-f004:**
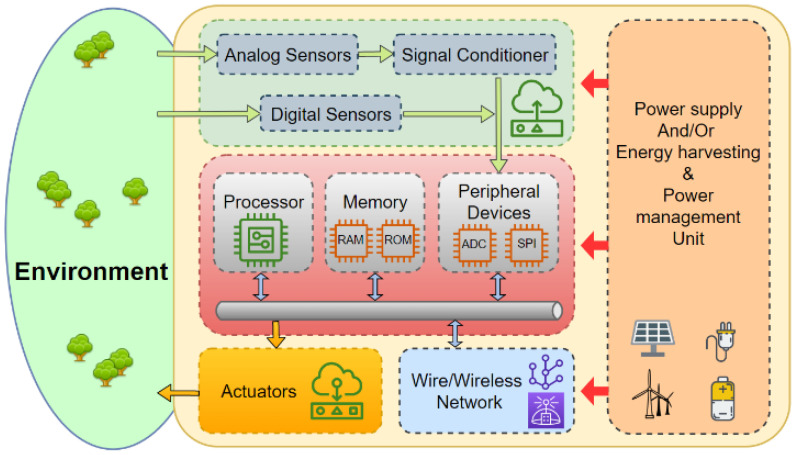
Basic hardware architecture of AIoT/IIoT/IoT devices.

**Figure 5 sensors-23-05074-f005:**
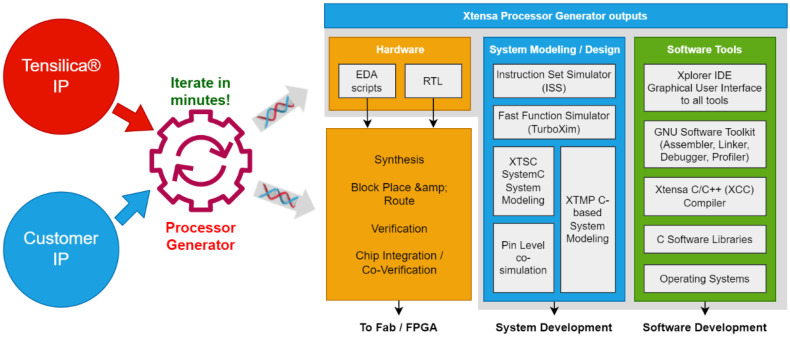
Cadence multi-IP co-design tools.

**Figure 6 sensors-23-05074-f006:**
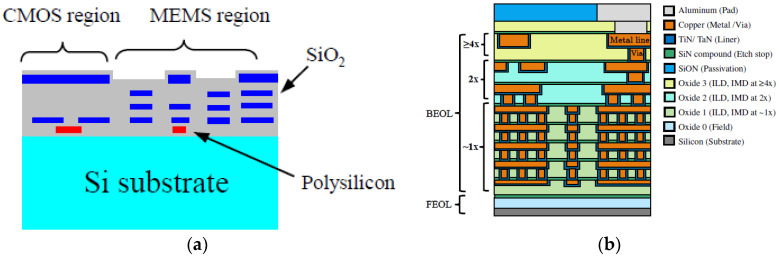
(**a**) MEMS and CMOS in the same substrate [47]; (**b**) MEMS/NEMS compatible CMOS switch [48].

**Figure 7 sensors-23-05074-f007:**
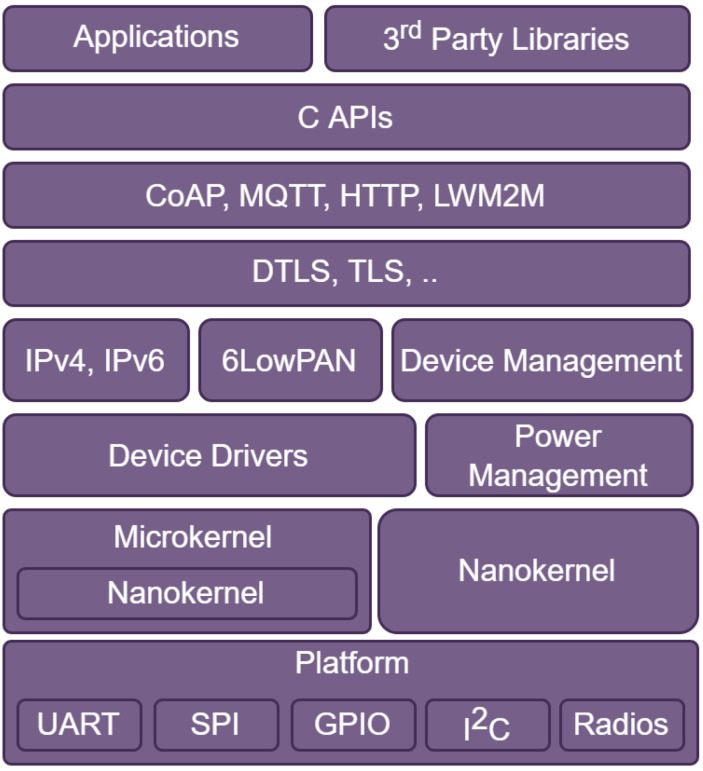
Zephyr firmware structure [52].

**Figure 8 sensors-23-05074-f008:**
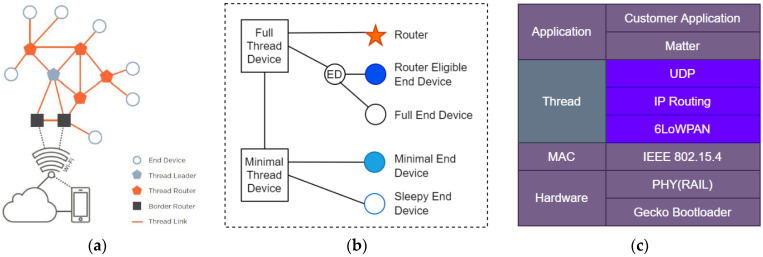
(**a**) Basic Thread network topology; (**b**) OpenThread devices; (**c**) matter application layer.

**Figure 9 sensors-23-05074-f009:**
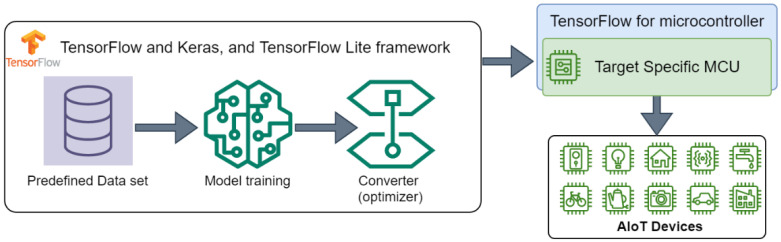
TensorFlow framework for microcontrollers.

**Figure 10 sensors-23-05074-f010:**
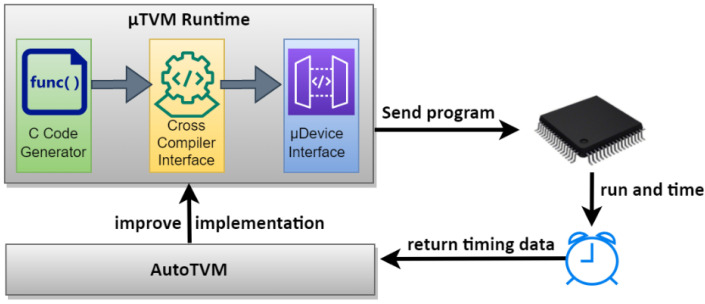
µTVM toolchain [63].

**Figure 11 sensors-23-05074-f011:**
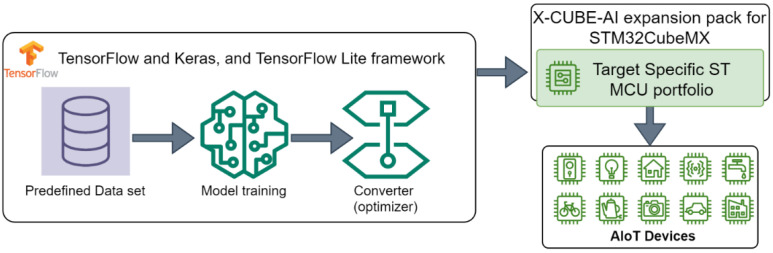
Deep learning development workflow to target the ST MCU portfolio.

**Figure 12 sensors-23-05074-f012:**
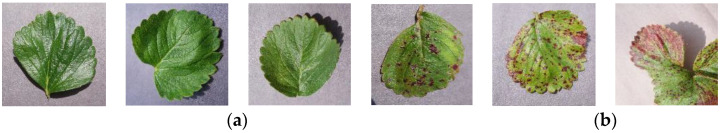
(**a**) Healthy leaves; (**b**) scorched leaves.

**Figure 13 sensors-23-05074-f013:**
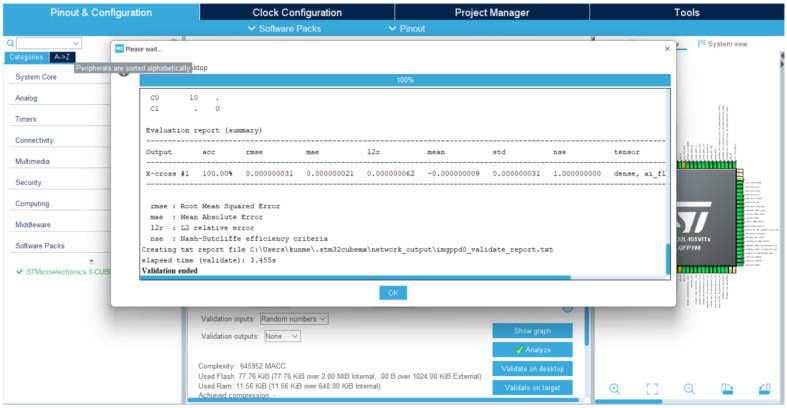
MCU resource requirement validated using the X-CUBE-AI expansion pack of STM32CubeMX.

**Figure 14 sensors-23-05074-f014:**
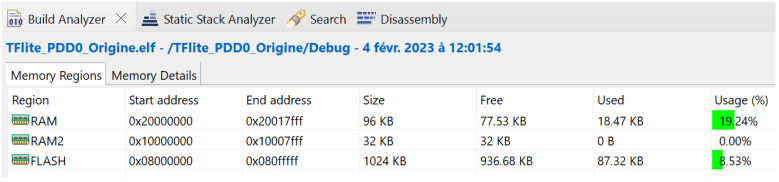
Application memory code sizes generated by STM32CubeIDE.

**Figure 15 sensors-23-05074-f015:**
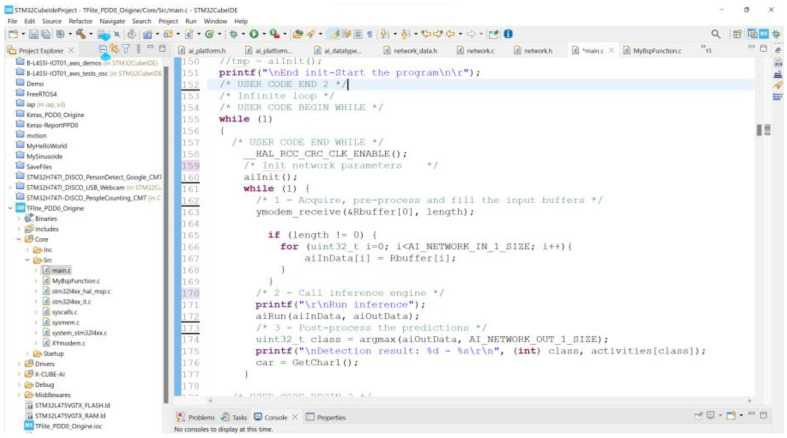
C code model displayed on the STM32CubeIDE screen.

**Figure 16 sensors-23-05074-f016:**
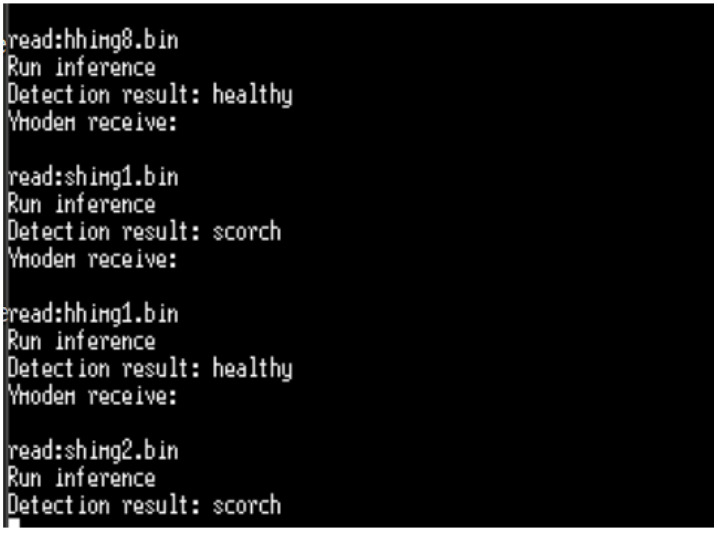
Results of embedded inference running on the STL32L4S5i discovery kit (screen copy).

**Figure 17 sensors-23-05074-f017:**
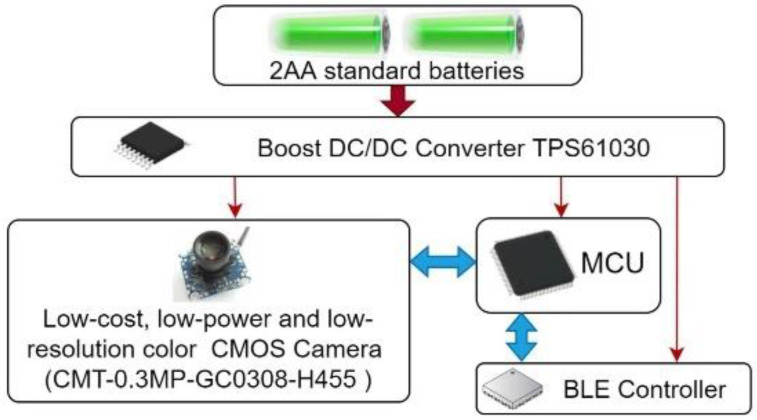
Bloc diagram of a low-power and low-cost BLE AIoT device dedicated to plant disease detection.

**Figure 18 sensors-23-05074-f018:**
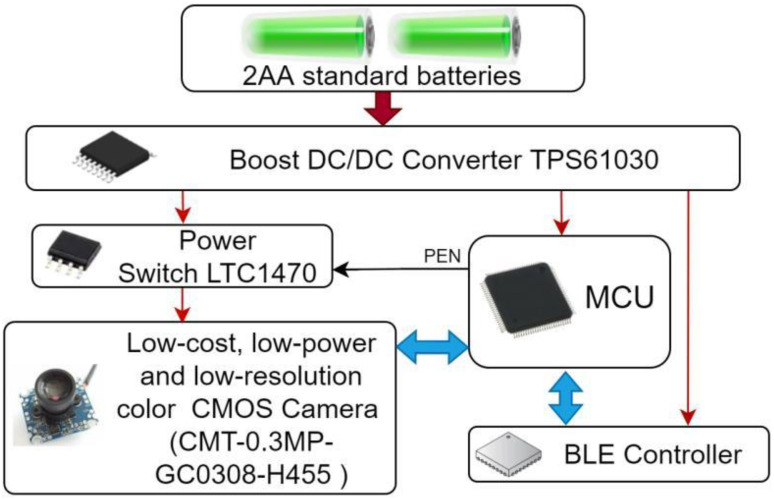
AIoT device integrating analog switch.

**Table 1 sensors-23-05074-t001:** Different types of wireless access media.

Item	Data Rate (Mbps)	Range (m)	Frequency (MHz)	Tx (mA)	Standard
Wi-Fi 1-3 [35]	2–54	<200	2400–5000	312	IEEE802.11 a/b/g
Wi-Fi 4 [36]	300–600	<200	2400/5000	312	IEEE802.11n
Wi-Fi 5 [37]	6900	<200	5000	610	IEEE802.11ac
Wi-Fi 6 [37]	9600	<200	2400 and 5000	NA	IEEE802.11ax
ZigBee [38]	0.02–0.25	<100	400–2400	25.8	IEEE802.15.4
BLE [39]	1–2	<10	2400	5.8	Bluetooth SIG
WMBUS [40]	0.0048–0.032768–0.0100	<500	433–868	37	OMS
Wi-Fi Halow [41]	0.15–3.9	<1000	863–930	8.5	IEEE802.11ah
WMBUS [42]	0.0024–0.0048–0.0192	LPWAN	169	703	OMS
LoRa [43]	0.300	LPWAN	433–868–915	20–120	LoRa Alliance
SP-L2 SigFox [44]	0.0001–0.500	LPWAN	413–1055	32	SigFox’s proprietary protocol
NB-IoT(5G)	0.10(DL)–0.02(UL)	LPWAN	700–800–900	167	3GPP
LTE-M	1	LPWAN	699–2200	202	3GPP

**Table 2 sensors-23-05074-t002:** Sensor classification based on sensor stimulus [45].

Stimulus	Measurand or Attribute
Mechanical	Pressure, flow rate, vibration, distance, velocity, acceleration, force, etc.
Thermal	Temperature, heat, heat flux, heat capacity, thermal conductivity, etc.
Electrical	Voltage, current, electric field, charge, resistance, capacitance, etc.
Magnetic	Magnetic flux, magnetic field, magnetic moment, etc.
Radiation	Infrared, X-rays, visible, ultrasonic, acoustic, radio waves, etc.
Chemical	pH, Ion, concentration, moisture, gas, oxygen, etc.
Biological	Taste, odor, protein, glucose, hormone, enzyme, microbe, etc.

**Table 3 sensors-23-05074-t003:** Accuracy results according to the CNN shape.

Shape/Accuracy	Baseline Keras Model (float32)	Float16 TFlite	Int8 TFlite	DQ TFlite
512 > 64 > 2	99.2	99.0	85.7	92.9
256 > 128 > 128 > 2	95.2	95.2	94.3	96.1
256 > 128 > 64 > 2	94.8	94.8	88.4	92.4
256 > 64 > 64 > 2	95.8	95.8	91.4	93.1
256 > 64 > 32 > 2	96.7	96.7	96.2	97.1
128 > 128 > 64 > 2	96.4	96.4	93.8	95.7
128 > 64 > 64 > 2	97.4	97.3	96.2	97.
128 > 64 > 32 > 2	97.9	97.9	96.4	98.3
64 > 64 > 32 > 2	93.1	93.1	88.1	92.1
64 > 32 > 32 > 2	95.8	95.7	92	96.4

**Table 4 sensors-23-05074-t004:** Power consumption of the ST and Ambiq portfolios used to implement an AIoT device in continuous active mode.

Item	µA/1 MHz	Current (mA)	Supply Voltage (V)	Active Max Frequency (MHz)	Power Consumption per Component (mW)
STM32L4S5VITx MCU [75]	110	-	3	120	39.60
ST BLE BlueNRG-MS [76]	-	8.2	3	-	24.6
Apollo4 MCU [77]	5	-	2.2	120	1.32
Apollo4 BLE [78]		3	3		9
CMOS camera	-	-	-	-	70

**Table 5 sensors-23-05074-t005:** Lifetime and power consumption ratio compared with the HPZ240 tower PC.

Item	Maximum Power Consumption (mW)	Ratio	AIoT Device Lifetime (2 Standard AA Battery 1.5 V, 2500 mAh)
PC server	65,000	1	-
AIoT based on ST portfolio	134.2	484	50.6 h (~2.1 days)
AIoT based on Ambiq portfolio	80.32	809	87 h (~3.6 days)

**Table 6 sensors-23-05074-t006:** Power consumption of the ST and Ambiq portfolios used to implement an AIoT device in sleep mode.

Item	Sleep Mode Current (µA)	Supply Voltage (V)	Power Consumption per Component (µW)	NB
STM32L4S5VITx MCU [75]	0.645	3.3	2.13	RTC clocked by LES quartz in low drive mode
ST BLE (BlueNRG-MS) [76]	1.7	3.3	5.61	
Apollo4 MCU [77]	7.7	3.3	25.41	deep sleep mode 3 and 3.3 V
Apollo4 BLE [78]	0.5	3.3	1.65	
CMOS camera switch quiescent current	1	3.3	3.3	

## Data Availability

Not applicable.

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
