# Peer review of "Trends and Challenges in AIoT/IIoT/IoT Implementation"

_sensors, 2023, doi:10.3390/s23115074_

Round 1

Reviewer 1 Report

Smart everything is on the way driving by the development of digital twin, metaverse, Industry 4.0 and autonomous vehicle and enjoys an incredible synergy of advancements in its core hardware and software technologies such as firmware (Zephyr and OpenThread), deep learning for microcontroller tools (e.g., X-CUBE-AI), VLSI and MCU (ultra-low power subthreshold and new memory technologies, new asymmetric multicore RISC-V architecture), MEMS/NEMS (fusion of MEMS/NEMS and CMOS processes), and wireless technologies (e.g., low energy high bandwidth and medium range Wi-Fi Halow). These advances enable the implementation of low-cost, low-power AIoT devices for Industry 4.0, smart home The paper is organized well. However, there are some points to be considered.

1.  Abstract needs to be more precise highlighting major contributions.

2.  It would be nice to explicitly list the future research directions. Please also discuss some limitations of your proposed method.

3. Please introduce some representative studies of novel approach of artificial general intelligence in the Introduction Section. These researches are critical studies in the field of brain-inspired intelligence to realize high-level intelligence, high accuracy, high robustness, and low power consumption in comparison with the state-of-the-art artificial intelligence works. These researches include: Robust Spike-Based Continual Meta-Learning Improved by Restricted Minimum Error Entropy Criterion; Heterogeneous Ensemble-based Spike-driven Few-shot Online Learning; SAM: A Unified Self-Adaptive Multicompartmental Spiking Neuron Model for Learning with Working Memory; Neuromorphic context-dependent learning framework with fault-tolerant spike routing.

4. The background of the proposed study should be further explained in detail. Some concepts are hard to comprehend without explaining clearly.

5. Grammar is expected to be further improved. Please check the manuscript carefully to remove the typos, improve the language and format.

6. Please discuss the hardware implementation of the proposed platform. Some neuromorphic computing researches should be discussed, because neuromorphic computing is a novel computing paradigm with low power consumption and high-speed response, including: Scalable digital neuromorphic architecture for large-scale biophysically meaningful neural network with multi-compartment neurons; Smart Traffic Navigation System for Fault-Tolerant Edge Computing of Internet of Vehicle in Intelligent Transportation Gateway;CerebelluMorphic: large-scale neuromorphic model and architecture for supervised motor learning; BiCoSS: toward large-scale cognition brain with multigranular neuromorphic architecture.

Author Response

Dear Reviewer,

Thanks a lot for your remarks and questions. The new version of our article is updated according to your remarks (green characters). 

Regards,

Reviewer 2 Report

On formatting and presentation: the paper is difficult to read due to many "Referencing Errors", which have to be corrected.  There were many explanations written in brackets with trailing  dots (...) when these can be changed to the terms "etc".  There were grammatical errors such as missing articles, conjunctions and incorrect usage of nouns such as Artificial Intelligent which should be Artificial Intelligence, plural and singular forms.

On contents: the paper title is stated as "Trends and Challenges in Safety ...".  However,  I see that the paper is lacking the discussion on the research gap about safety in the AIoT etc, and how important safety itself is to be tackled for AIoT.  Towards the end of the paper, no discussions on how "Safety" are incorporated to the work of detecting disease in strawberry.   And what makes detection of disease in strawberry requiring the safety criteria?

It would be great to know what are the final hyperparameters used to get the final results of the detection.

I attached an annotated file containing the typo, grammatical errors.

Author Response

Dear reviewer,

Thanks a lot for your remarks and suggestions. The new version of our article is updated (green characters) by considering your advices.

Regards,

Round 2

Reviewer 1 Report

I am satisfied with the revision. It can be accepted if the authors can consider the following points:

Brain-inspired intelligence can realize high-level intelligence, high accuracy, high robustness, and low power consumption in comparison with the state-of-the-art artificial intelligence works. These researches include: Robust Spike-Based Continual Meta-Learning Improved by Restricted Minimum Error Entropy Criterion; Heterogeneous Ensemble-based Spike-driven Few-shot Online Learning; SAM: A Unified Self-Adaptive Multicompartmental Spiking Neuron Model for Learning with Working Memory; Neuromorphic context-dependent learning framework with fault- tolerant spike routing.

Please have a discussion considering these SOTA researches.

Author Response

We have added the following text based on your suggestion:

The fourth FPGA-based neuromorphic platforms outperform previous platforms in terms of power consumption, response time and number of neurons implemented. Yang et al. proposed a CerebelluMorphic system using six Intel Stratix III EP3SL340 FPGAs to realize the large-scale neuromorphic cerebellar network with approximately 3.5 million neurons and 218.3 million synapses [24]. While Wang et al. present a new abstraction of a neuromorphic architecture into clusters represented by minicolumns and hypercolumns, analogously to the fundamental structural units observed in neurobiology. As a result, an implementation on one Altera Stratix V FPGA was able to simulate 20 million to 2.6 billion leaky-integrate-and-fire (LIF) neurons in real time [25]. Intel Programmable Solutions Group (neuromorphic computing) and the International Center for Neuromorphic Systems at Western Sidney University (WSU) are building a neuromorphic platform using 168 Intel Stratix 10 PGAs with high-bandwidth memory (HBM) and an accelerator configurable network protocol (COPA) to simulate the human cortex (LIF model).It is estimated that human cerebral cortex has  10 to 20 billion neurons and 60 to 240 trillion synapses [26]. Proofs of concepts (PoCs) developed on FPGA-based neuromorphic and memristor IMC platforms will lead to the next significant advances in SoC design of low-cost, low-power AIoT devices.

All of your article proposals are very interesting, but we have cited one that seemed important to us and that corresponds to the outline of our article. Note that our work focuses on the ecosystem to implement low-cost, low-power AIoT devices.

We believe that in the coming years, advances in SNNs, memristor materials, and NEMS/SEMS sensors will dramatically improve the performance of AIoT devices.

Thanks a lot for your suggestions.

Reviewer 2 Report

Comments have been addressed except please check typo at Line Manuscript has improved at Line 537 & Line 667, when a singular form of leaves is used, it should be "leaf".  So Line 537 & Line 667, should be "...scorch leaf...".  If wish to maintain the term as "leave", a plural form is supposed to be used, which is "leaves".

Author Response

Dear rewiewer,

We correct the grammatical errors you indicated. Thanks a lot for your help.
